# Resistance to Resveratrol Treatment in Experimental PTSD Is Associated with Abnormalities in Hepatic Metabolism of Glucocorticoids

**DOI:** 10.3390/ijms24119333

**Published:** 2023-05-26

**Authors:** Vadim E. Tseilikman, Julia O. Fedotova, Olga B. Tseilikman, Jurica Novak, Marina N. Karpenko, Victoria A. Maistrenko, Svetlana S. Lazuko, Lyudmila E. Belyeva, Mustapha Kamel, Alexey V. Buhler, Elena G. Kovaleva

**Affiliations:** 1Scientific and Educational Center ‘Biomedical Technologies’, School of Medical Biology, South Ural State University, 454080 Chelyabinsk, Russia; vadimed@yandex.ru (V.E.T.); diol2008@yandex.ru (O.B.T.); 2Laboratory of Neuroendocrinology, I.P. Pavlov Institute of Physiology RAS, 6 Emb. Makarova, 199034 Saint Petersburg, Russia; julia.fedotova@mail.ru; 3Faculty of Fundamental Medicine, Chelyabinsk State University, 454001 Chelyabinsk, Russia; 4Department of Biotechnology, University of Rijeka, 51000 Rijeka, Croatia; 5Center for Artificial Intelligence and Cyber Security, University of Rijeka, 51000 Rijeka, Croatia; 6Pavlov Department of Physiology, Institute of Experimental Medicine, 197376 Saint Petersburg, Russia; mnkarpenko@mail.ru (M.N.K.); sch_viktoriya@mail.ru (V.A.M.); 7Department of Physiology, Vitebsk State Medical University, Frunze Av. 27, 210023 Vitebsk, Belarus; lazuko71@mail.ru; 8Department of Pathophysiology, Vitebsk State Medical University, Frunze Av. 27, 210023 Vitebsk, Belarus; lyudm.belyeva2013@yandex.ru; 9Research, Educational and Innovative Center of Chemical and Pharmaceutical Technologies Chemical Technology Institute, Ural Federal University Named after the First President of Russia B. N. Yeltsin, 620002 Ekaterinburg, Russia; mustapha.mohaab@gmail.com (M.K.); zellist@mail.ru (A.V.B.)

**Keywords:** PTSD, *trans*-resveratrol, glucocorticoids, anxiety, CYP3A, 11-β-hydroxysteroid dehydrogenase type 1, molecular dynamics

## Abstract

Glucocorticoids are metabolized by the CYP3A isoform of cytochrome P450 and by 11-β-hydroxysteroid dehydrogenase type 1 (11β-HSD-1). Experimental data suggest that post-traumatic stress disorder (PTSD) is associated with an increase in hepatic 11β-HSD-1 activity and a concomitant decrease in hepatic CYP3A activity. *Trans*-resveratrol, a natural polyphenol, has been extensively studied for its antipsychiatric properties. Recently, protective effects of *trans*-resveratrol were found in relation to PTSD. Treatment of PTSD rats with *trans*-resveratrol allowed the rats to be divided into two phenotypes. The first phenotype is treatment-sensitive rats (TSR), and the second phenotype is treatment-resistant rats (TRRs). In TSR rats, *trans*-resveratrol ameliorated anxiety-like behavior and reversed plasma corticosterone concentration abnormalities. In contrast, in TRR rats, *trans*-resveratrol aggravated anxiety-like behavior and decreased plasma corticosterone concentration. In TSR rats, hepatic 11β-HSD-1 activity was suppressed, with a concomitant increase in CYP3A activity. In TRR rats, the activities of both enzymes were suppressed. Thus, the resistance of PTSD rats to *trans*-resveratrol treatment is associated with abnormalities in hepatic metabolism of glucocorticoids. The free energy of binding of resveratrol, cortisol, and corticosterone to the human CYP3A protein was determined using the molecular mechanics Poisson–Boltzmann surface area approach, indicating that resveratrol could affect CYP3A activity.

## 1. Introduction

Post-traumatic stress disorder (PTSD) is a disorder resulting from severe and life-threatening psychological trauma, such as exposure to war, a terrorist attack, a traffic accident, or a natural disaster. PTSD is a relatively common disorder and is found in 2–5% of the general population [1] while in people who have experienced severe stress and anxiety, PTSD symptoms are found in as many as 45–70%. The prevalence of PTSD has been found to depend on the type of injury, gender, age, and sociocultural characteristics. PTSD is often complicated by depression, heart attack, stroke, gastric ulcers, diabetes mellitus, Alzheimer’s disease, and even cancer [2]. PTSD patients are very often resistant to treatment [3]. In particular, up to 40% of PTSD patients are resistant to drug therapy [3,4]. There are studies on the effect of steroid medications, propranolol, omega-3 fatty acids, anticonvulsants such as gabapentin, selective serotonin reuptake inhibitors (SSRIs), or 5-hydroxytryptophan as agents to prevent post-traumatic stress disorder [5]. Selective serotonin reuptake inhibitors are the first-line medications for the treatment of PTSD. However, these drugs are characterized by a variety of side effects, including an increased risk of developing hemorrhagic shock and serotonin syndrome. [6]. Therefore, the search for new approaches to treat PTSD is very important.

In this context, our attention has been drawn to resveratrol, known as 3,5,4-trihydroxy-*trans*-stilbene (Figure 1), which is a type of biological polyphenol obtained mainly from peanuts, grapes, and mulberries [7]. Due to its strong antioxidant action, resveratrol is effective in cardiovascular diseases, type 2 diabetes mellitus, immunodeficiency states, oncology, and behavioral disorders [8]. The numerous beneficial biological effects of resveratrol are due to its multiple molecular targets [9]. Crucially, both resveratrol and SSRIs share a common target, the serotonin transporter. Thus, in an experimental model of PTSD, resveratrol was shown to be non-inferior to SSRIs in efficacy in correcting anxiety disorders in PTSD. Anxiety disorders in PTSD, in turn, are induced by the process of time-dependent sensitization (TDS), which is a classic animal model for mimicking post-traumatic stress disorder (PTSD) [10]. In addition, TDS is associated with abnormalities in the limbic–hypothalamic–pituitary adrenal (LHPA) system. Resveratrol reversed the abnormalities of the LHPA characteristic of PTSD, both by increasing the production of corticoliberin and by regulating tissue glucocorticoid (GC) metabolism. GC is metabolized by 11-beta-hydroxysteroid dehydrogenase types 1 or 2 (11β-HSD-1 or 11β-HSD-2) and isoforms of cytochrome P450 belonging to the CYP3A subfamily. It has been shown that 11β-HSD-1-dependent hepatic metabolism of glucocorticoids predominates after PTSD. In contrast, the CYP3A-dependent metabolic pathway is suppressed under PTSD. These data agree well with in silico data indicating the ability of resveratrol to interact with 11β-HSD-1 [11]. Molecular docking data indicate the ability of resveratrol to inhibit the activity of CYP3A isoforms [12]. In contrast to 11β-HSD-1, CYP3A enzymes have a wide range of substrate specificities. In addition to glucocorticoids, these enzymes also metabolize various xenobiotics [13]. Unfortunately, the effect of resveratrol on the catalytic activity of CYP3A isoforms toward glucocorticoids has not been investigated in in silico studies. In addition, the effect of resveratrol on the relationship between the 11β-HSD-1 and CYP3A pathways of glucocorticoid metabolism in PTSD has not been investigated.

## 2. Results

### 2.1. Effects of Trans-Resveratrol Treatment on EPM Test Scores in PTSD Rats

We exposed rats to predator stress for 10 days. After 14 days of recovery, we performed an elevated-plus-maze (EPM) test. EPM test values are collected in Table 1. PTSD rats showed a significant decrease in time spent in the open arms and an increase in time spent in the closed arms. Based on the anxiety index score (AI), *trans*-resveratrol-treated rats were classified into two phenotypes. The first phenotype was defined as treatment-susceptible rats (TSR; AI < 0.8), whereas the second phenotype was defined as treatment-resistant rats (TRR; AI > 0.8).

In TSR rats, treatment with *trans*-resveratrol at a dose of 40 mg/kg abolished all behavioral effects of PTSD. Time spent in open arms was 236% higher (*p* = 0.019) and time spent in closed arms was 20% lower (*p* = 0.019) than in PTSD rats. In contrast, *trans*-resveratrol treatment failed to correct the anxiety-like behavior of PTSD in TRR rats. There were significant differences in EPM test scores between TRR and TSR rats. Specifically, time spent in the closed arms was 32% higher (*p* = 0.019) and time spent in the open arms was 80% (*p* = 0.019) lower than TSR rats. Overall, the data presented in Table 1 show that the behavioral patterns of PTSD rats and TRR rats were similar to each other.

### 2.2. Effects of Trans-Resveratrol Treatment on Plasma Concentrations of Corticosterone in Rats with PTSD

It was shown (Figure 1) that plasma corticosterone (CNE) concentration was 20% lower (*p* = 0.028) in PTSD rats than in control rats. There was a negative correlation between the AI score and plasma CNE concentration (r = −0.62; *p* < 0.05). In TSR rats, plasma CNE concentration was 51% higher than in PTSD rats (*p* = 0.00024). There were no significant differences in plasma corticosterone concentration between the TRR rats and the PTSD rats (*p* = 0.96). In the TSR rats, plasma corticosterone concentration was 25% higher (*p* = 0.022) than in the control group.

### 2.3. Effects of Trans-Resveratrol Treatment on Liver 11β-HSD-1 Activity

In the PTSD rats, hepatic 11β-HSD-1 activity was 223% higher (*p* = 0.00018) than in the control group (Figure 2). There was a negative correlation between plasma CNE concentration and hepatic 11β-HSD-1 activity (r = −0.58; *p* < 0.05). In TSR rats, hepatic 11β-HSD-1 activity was 63% lower (*p* = 0.00017) than in PTSD rats and in TRR rats it was 36% lower (*p* = 0.019) than in the PTSD group.

### 2.4. Effects of Trans-Resveratrol Treatment on CYP3A Activity in the Liver

The data presented in Figure 2 show that CYP3A activity in the liver of PTSD rats was 50% (*p* = 0.007) lower than in the control group. In the PTSD group, a positive correlation was observed between CYP3A activity and CNE concentration (r = 0.63; *p* < 0.05). In TSR rats, hepatic CYP3A activity was 140% higher (*p* = 0.019) than in PTSD rats and 220% higher (*p* = 0.00018) than in the TRR group. Of note, a positive correlation was also observed between CYP3A activity and CNE concentration in TSR rats (r = 0.54; *p* < 0.05). In the TRP rats, CYP3A activity was 34% lower than in the PTSD group (*p* = 0.033).

### 2.5. Molecular Docking and Molecular Dynamics Simulations

The docking of *trans*-resveratrol (RES) to the binding pocket of CPY3A4 isoform (CYP) gives an estimate of the free energy of binding of resveratrol to CYP of −8.1 kcal mol^−1^. Both cortisol (COL) and CNE have better docking scores then resveratrol, with −9.0 kcal mol^−1^ and −8.9 kcal mol^−1^, respectively. These figures should be taken with caution because the structure of the enzyme was frozen during the experiment, which is one of the drawbacks of the approach [14]. To overcome this obstacle, molecular dynamics (MDs) simulations of RES, COL, and CNE bound in the active pocket of CYP were performed with the aim of investigating the stability of the RES:CYP, COL:CYP, and CNE:CYP complexes and calculating the binding free energy with higher accuracy. The docked structures were used as the initial geometry of the complexes for the subsequent MD simulations. In all docked structures, the ligands were located near the prosthetic heme group. The hydrogen from the hydroxyl group of the benzene-1,3-diol moiety of RES is only 3.13 Å away from the centroid of heme (HEM), and the angle between the planes is 32°. Analysis of the docked structure with PLIP [15] showed that hydrogen bonds as well as hydrophobic interactions are important for the bonding.

To better understand the interaction pattern between ligand and enzyme and to investigate the conformational dynamics and stability of the RES:CYP complex, a 1 μs long molecular dynamics simulation was performed. The root-mean-square deviations (RMSDs) and the radius of gyration (RoG) of the complex were calculated against the initial structure to monitor the stability of the complex (Figure 3). The system reached equilibrium in 70 ns, with the average RMSD fluctuating around 2.47 ± 0.18 Å. Cluster analysis (vide infra) indicated conformational changes of the protein around 70 ns and at 420 ns. When the RMSD values are divided into two time periods (from the beginning of the simulation to 70 ns and from 70 ns to the end) and the mean values are calculated, it is seen that the RMSD for the first period is 2.00 ± 0.29 Å and for the second 2.51 ± 0.10 Å. Plotting the radius of gyration against the simulation time confirms the convergence of the trajectory and the stability of the complex. As a measure of the distribution of atoms in the complex, the RoG is a valuable descriptor describing the compactness of the complex. The RoG mean value is 22.9 ± 0.2 Å, and the division of the RoG values into two segments (analogous to the RMSDs) shows no significant differences.

The COL:CYP and CNE:CYP complexes are stable on the μs time scale. RMSD values are low, although a little higher than for RES:CYP. On the other hand, the mean RoG values are lower, indicating greater compactness of the complexes (Appendix A).

The residual fluctuations of CYP3A4 protein in the complex were analyzed after calculating the root-mean-square fluctuations (RMSFs) for each residue (Figure 2) using the following formula
(1)$$RMSF_{i}={\left[\frac{1}{t}\overset{t}{\underset{t_{j}=1}{\sum }}{\left|r_{i}\left(t_{j}\right)-r_{i}^{ref}\right|}^{2}\right]}^{1/2}$$
where *t* is the time over which the average is calculated and r*_i_* is the position of particle *i.* RMSFs are calculated relative to the reference structure, which in our case was the average structure. High RMSF values indicate stronger oscillations of the residues during the simulation. The average RMSF value for the RES:CYP complex is 1.41 Å, indicating that the secondary structure of the protein is well conserved. However, neglecting the higher flexibility of residues at the *C*-terminus, there are three regions with an RMSF value greater than 3.0 Å. The first region comprises the residues between Ser52 and Lys55, part of the H2 helix (Appendix A). The second flexible region (Arg260-Thr264) is part of a 13-residue-long unstructured loop connecting α-helixes H14 and H15. This motif is located on the surface of the protein, in direct contact with water, just like the most flexible loop (Lys282-Ser286) between H15 and H16 helices. The conservation of the secondary structure is confirmed by calculating the secondary structure along the trajectory using the DSSP algorithm [16] (Appendix A).

When COL or CNE are bound in the catalytic pocket, the flexibility of the residues of CYP is increased (Appendix A). The effect is more pronounced with CNE. The difference is greatest for residues forming the α-helices from H9 to H12 and the loops connecting them (Appendix A). The most flexible residues (with an RMSF greater than 3.0 Å) are part of the loop connecting helices H15 and H16, with Thr284 having a maximum value of 6.2 Å. In contrast to the RES:CYP complex, residues in direct contact with CNE (Leu211-Phe213) also exhibit increased flexibility in the CNE:CYP complex.

RES is a type of natural phenol with one hydroxyphenyl and one dihydroxyphenyl component that obeys Lipinski’s rule of five. Due to its structure, it can form multiple hydrogen bonds, both as a hydrogen donor and a hydrogen acceptor. In addition, aromatic interactions could play an important role in its positioning within the active pocket. The catalytic pocket of CYP is hidden from the solvent. It contains several residues that can form hydrogen bonds, such as Arg105, Ser119, Arg212, and Thr309. The time evolution of the number of hydrogen bonds formed is shown in Figure 4. While the number of hydrogen bonds between RES and CYP varies between 0 and 7, the average number is 1.3. On average, both COL and CNE form more hydrogen bonds than RES, with 1.7 and 2.2, respectively (Appendix A). One of the CNE hydroxy groups is a hydrogen donor for the carboxyl group of HEM, while the other hydroxy group forms a hydrogen bond with the side chain of Glu 374.

### 2.6. Identification of Three Conformations

The *k*-means cluster analysis based on the RMSD of Cα atoms was used to identify relevant conformations of CYP. We tested all *k* values between 2 and 10, and to find the optimal number of conformations and the results were analyzed using the Davies–Bouldin index (DBI), the pseudo-F statistic (pSF), and the ratio of sum of squares of the regression and the sum of squares error (SSR/SST) (Appendix A). The existence of three relevant conformations was confirmed (Table 2). The structure closest to the centroid of the cluster was chosen as the representative structure of each conformation. The RMSD relative to the first structure from the MD simulation with the position of the representative conformations is shown in Figure 3. In Figure 5, the structures of three representative conformations are superimposed. Conformation C (RES:CYP-C), the least populated, is present only at the beginning of the trajectory. The main difference from conformations A (RES:CYP-A) and B (RES:CYP-B) is the flexible loop connecting α-helixes H14 and H15 (L1 in Figure 5). The loop connecting helices H15 and H16 (L2 in Figure 5) is the main difference between RES:CYP-A and RES:CYP-B. The residues of the L1 and L2 loops were identified by RMSF analysis as the most flexible, with RMSF values greater than 3 Å. As expected, the secondary structure was well preserved throughout the simulation.

In addition to the changes in the CYP protein itself, a reorganization of the intermolecular interaction between RES and CYP in the catalytic pocket was also observed. Leu483 forms a hydrogen bond with hydroxyphenyl in RES:CYP-B and RES:CYP-C, with an average bond length of 1.79 Å. The bond is present during 32% of the simulation time, which is only slightly shorter than the sum of the population of the two conformations. While only Ile369 and Leu482 form hydrophobic interactions with RES in RES:CYP-C, additional interactions are formed with Phe304 and Ala370 in RES:CYP-B. The interaction pattern in RES:CYP-A is completely different, indicating a significant reorientation of RES within the catalytic cavity (Figure 6). Three residues are involved in hydrogen bonds (Thr309, Glu374, and Arg375), and Ala370 and Glu374 in hydrophobic interactions, while Arg105 forms a π-cation interaction. HEM also plays an important role in ligand positioning—in addition to hydrophobic interactions, it forms a 1.67 Å long hydrogen bond with one of the hydroxyl groups of the dihydroxyphenyl moiety.

The dynamics cross-correlation map (DCCM) was calculated to identify correlated motions of the CYP residues. The positive values represent positively correlated motions, while negative values represent anti-correlated motions. Since the values of the DCCM matrix are close to 0, it can be assumed that there are no significant (anti-)correlated movements (Appendix A).

### 2.7. Binding Free Energy Calculation

The molecular mechanics/generalized Born surface area (MM/GBSA) approach (Equations (3)–(5)) was used to estimate the free energy of binding of RES to CYP. The binding energy, without unfavorable entropic (S) contribution, is −21.4 ± 2.4 kcal mol^−1^. If -TΔS is included, total binding free energy at 310 K is –2.6 kcal mol^−1^. More detailed analysis revealed that van der Waals interactions (−30.8 ± 1.3 kcal mol^−1^), together with electrostatic (−18.3 ± 3.6 kcal mol^−1^) and nonpolar solvation energy (−4.1 ± 0.2 kcal mol^−1^), play the major role in binding. Although hydrophobic interactions between RES and CYP make a crucial contribution, intermolecular hydrogen bonding should not be neglected. In our previous studies of the main protease of SARS-CoV-2 virus [17] and the NS3 protease of Kyasanur forest disease virus [18], a threshold of −1.5 kcal mol^−1^ was set for the free energy of binding of a single residue to classify it as a residue with dominant contribution to binding. If the same criteria were applied in this study, only HEM (−3.2 kcal mol^−1^) would meet the chosen criteria. Lowering the threshold to −1.1 kcal mol^−1^, Ile369 (−1.3 kcal mol^−1^), Thr309 (−1.2 kcal mol^−1^), Leu482 (−1.2 kcal mol^−1^), and Arg105 (1.2 kcal mol^−1^) are identified as contributing significantly to binding. Heme and Thr309 are involved in hydrogen bonding with RES, both as hydrogen acceptors. Heme, Leu482, and Ile369 enhance binding via hydrophobic interactions (Figure 7). Our results quantify the qualitative results of the docking experiments of Basheer et al. [12,19,20].

Both COL (–31.0 ± 2.6 kcal mol^−1^) and CNE (–37.6 ± 2.7 kcal mol^−1^) have a more favorable free energy of binding to CYP than RES, and CNE binds most strongly. If entropy is included, free energy of binding at 310 K is –7.0 kcal mol^−1^ and –14.7 kcal mol^−1^ for COL and CNE, respectively. Looking at the contributions to the total free energy of binding, we find that the contribution of van der Waals interactions is lower by more than 10 kcal mol^−1^ for resveratrol compared with COL and CNE. At the same time, the electrostatic interactions and the nonpolar solvation energies are of a comparable magnitude (Table 3).

CNE has the highest contribution of electrostatic interactions, which may be related to residues Arg105 and Arg106 with protonated side chains. COL, in addition to Arg106, has Phe108 and Phe 215 as the residues contributing the most to the free energy of binding, with phenyl groups participating in favorable aromatic interactions [21].

## 3. Discussion

In this study, the anxiety index was used for the first time as a criterion for classifying rats pharmacologically treated with resveratrol into two phenotypes. For the first phenotype with AI < 0.8, *trans*-resveratrol therapy is quite effective, whereas the second phenotype with AI > 0.8 is characterized by resistance to pharmacological correction. For the first time, this index was successfully used as a cutoff criterion for classifying experimental animals into susceptible and resistant to experimental PTSD [22]. It is important to note that AI = 0.8 is considered critical based on discriminant analysis of a representative sample of experimental animals. At the same time, it was shown that interphenotypic differences between rats susceptible and resistant to PTSD were not limited to behavioral indicators [23]. For example, PTSD-susceptible rats were characterized by decreased levels of dopamine with concomitant decreases in the expression of MAO-A [24] and in brain-derived neurotrophin factor (BDNF) levels [24]. In addition, animals susceptible to PTSD are characterized by the presence of oxidative stress, damage to the myocardium, liver, and adrenal glands [23]. Differences between PTSD-susceptible and -resistant animals also involve neuroendocrine regulation and metabolic parameters. In particular, PTSD-susceptible rats were initially found to be characterized by increased 11β-HSD-1 activity in the liver, whereas PTSD-resistant animals were initially found to have increased CYP3A activity [25]. In view of these facts, we considered it appropriate to use the anxiety index to investigate the causes of the development of drug resistance in experimental PTSD. Our studies demonstrate the efficacy of this approach with respect to resveratrol, in which the proportion of animals sensitive to resveratrol (64%) significantly exceeded the proportion of resistant animals (36%).

Furthermore, this approach could be successfully applied to the analysis of other pharmacological drugs used to treat both PTSD and anxiety disorders in the future. In this study, we deepened the current understanding of the effect of resveratrol on tissue glucocorticoid metabolism in experimental PTSD. Resveratrol was shown to effectively inhibit both the 11β-HSD-1-dependent and CYP3A-dependent pathways of corticosterone and cortisol biotransformation. This fact motivates us to consider not only the protective effects of resveratrol but also its side effects. The protective effect of resveratrol against PTSD is explained by its ability to increase the production of neurotrophins in the brain and decrease the activity of 11β-HSD-1 in the liver. This is in good agreement with our previous findings that increased hepatic 11β-HSD-1 activity causes a decrease in blood corticosterone concentration [26]. Decreased glucocorticoid levels, in turn, cause behavioral disturbances and abnormalities in the metabolism of monoamine neurotransmitters in various brain structures [26]. Therefore, the inhibitory effect of resveratrol on hepatic 11β-HSD-1 activity could block the initial stages of PTSD development. However, the ability of resveratrol to inhibit the CYP3A-dependent pathway of glucocorticoid metabolism may be indicative of the side effects of resveratrol. The validity of this assumption is supported by the fact that TRR rats showed a decrease in CYP3A activity, even compared with PTSD rats. In contrast, CYP3A activity was increased in TSR rats compared with PTSD rats. At the same time, a greater decrease in enzyme activity related to hepatic 11β-HSD-1 was observed in TSR rats compared with TRR rats. Despite a decrease in 11β-HSD-1 and CYP3A activities, resveratrol administration did not restore plasma corticosterone levels in TRR rats. Protective effects of resveratrol on the adrenal glands have already been noted in experimental PTSD. They are probably characteristic of TSR rats and absent in TRR rats. This may explain the fact that corticosterone concentrations were higher in TSR rats than in control rats. It is important to note that CYP3A enzymes are inducible and their expression can be triggered by glucocorticoids. [13] This fact is in good agreement with the positive correlations between corticosterone concentration and the level of CYP3A activity in TSR rats and in PTSD rats. However, in TSR rats, a marked increase in corticosterone level was observed with a concomitant increase in CYP3A activity. In contrast, both indicators were reduced in PTSD rats. On the other hand, hepatic 11β-HSD-1 activity was greatly increased. It is possible that the decrease in corticosterone levels in PTSD is due to an increase in 11β-HSD-1 activity. Against the background of low corticosterone levels, the levels of proinflammatory cytokines such as IL-1, TNFα, and IL-6 increase [27]. These, in turn, have the ability to inhibit CYP3A activity. Increased levels of proinflammatory cytokines have already been found in experimental PTSD.

In silico methods were used to determine the free energy of binding of resveratrol to 11β-HSD-1. It seems clear that resveratrol is a weak competitive inhibitor of 11β-HSD-1 because its binding potential is quite low. Here, we performed an analogous computational study for the binding of resveratrol to CYP3A4 and compared the results for the binding of corticosterone and cortisol. Based on the calculation of the free energy of binding to the CYP3A4 isoform, the probability that resveratrol acts as a competitive inhibitor is not high. According to the available data, resveratrol is more effective as a competitive inhibitor against 11β-HSD-1 than against CYP3A. The possibility of allosteric modulation of 11β-HSD-1 and CYP3A activity by resveratrol remains open.

In general, the obtained results allow us to identify the mechanisms of the protective effect of resveratrol in experimental PTSD and to assess the risks of its side effects. The direct effects of resveratrol on tissue metabolism of glucocorticoids are evident in its inhibition of 11β-HSD-1 and CYP3A activities in the liver. In TRR rats, the in vivo data are in complete agreement with the computational data that resveratrol acts as an inhibitor of the enzymatic activities of both enzymes. However, the inhibitory effect of resveratrol on hepatic 11β-HSD-1 activity is more pronounced in TSR rats. Therefore, in contrast to TRR rats, complete recovery of corticosterone levels occurred in TSR rats, resulting in the induction of CYP3A enzymes.

The side effects of resveratrol at the behavioral level are presented. Before starting clinical trials, it is necessary to determine the specifics of the side effects of resveratrol in relation to the internal organs. First of all, it is necessary to clarify to what extent oxidative stress is involved in the development of side effects. All this requires additional experimental studies. Another problem is to find ways to eliminate the side effects of resveratrol. In this direction, it seems promising to us to administer resveratrol together with probiotics.

## 4. Materials and Methods

The animals were divided into the following groups:Control rats (treated with vehicle only for 10 days, *n* = 12);PTSD (rats previously exposed to chronic predator stress followed by a two-week rest, *n* = 12);RES + PTSD (an effective dose of resveratrol was administered to rats via a tube one hour before the onset of predatory stress; *n* = 22). The effective dose of resveratrol was determined based on data presented in [11]. After performing a behavioral test in the elevated-plus maze (EPM), animals were classified into two phenotypes based on AI value: treatment-sensitive Rats (TSR; AI < 0.8) based on AI value, whereas the second phenotype was defined as treatment-resistant rats (TRR; AI > 0.8).

To elicit PTSD, we used a modified predator stress model originally described by Cohen and Zohar [22], as used in our previous studies [28]. Predator stress was achieved by exposing rats in the PTSD groups to the odor of cat urine for 15 min daily for 10 days. Subsequently, the PTSD rats were given a 14-day rest period under stress-free conditions. On day 15, an EPM test was performed, and the animals were sacrificed. The rats were sacrificed with an overdose of diethyl ether and decapitated, and the blood was collected. Blood plasma and liver were frozen at −70 °C for biochemical studies.

### 4.1. Behavioral Assessment

On day 14 after the last PSS exposure, the anxiety level of the rats was measured using the elevated plus maze (EPM) test, as described previously. The test lasted ten minutes. AI was calculated using the following formula:(2)$$AI=1-\frac{1}{2}\left(\frac{T_{op}}{T}+\frac{N_{op}}{N}\right)$$
where *T_op_* is the time spent in the open arms, *T* is the total time in the maze, *N_op_* is the number of entrances to the open arms, and *N* is the total number of all entrances. The EPM test is one of the most commonly used tests to investigate anxiety-like behavior in rodents. The apparatus consisted of four branched arms (50 × 10 cm) with two open arms and two closed arms (40 cm high). The arms were connected by a central arm.

Plasma concentrations of CNE were determined using an enzyme-linked immunosorbent assay (ELISA) kit for measuring CNE (Cusabio ELISA Kit, Houston, TX, USA) according to the manufacturer’s instructions. The sensitivity of the assay was 0.25 ng/mL, and the coefficients of variation within and between assays were <5%.

Hepatic 11β-HSD-1 activity was assessed by a decrease of 10 μM corticosterone (Sigma Aldrich Ltd., Saint Louis, MO, USA). A total of 0.1 M sodium phosphate buffer (pH 8.5) containing 1.5 mM NADP (Sigma Aldrich Ltd., Saint Louis, MO, USA) was used. The samples were incubated at 37 °C for 60 min. The sample containing the substrate (corticosterone) was added at the end of the incubation, and the blank sample containing an equivalent volume of solvent was incubated simultaneously. The changes in fluorescence intensity (405 nm excitation wavelength and 546 nm emission wavelength) were measured using a VERSA FLUOR spectrofluorometer (Bio-Rad, Hercules, CA, USA).

Hepatic CYP3A activities were assessed as previously described [26]. Briefly, livers were homogenized in 1.15% KCl. Homogenates were centrifuged at 9000× *g* for 20 min, followed by centrifugation of the supernatant at 100,000× *g* for 60 min. The microsomal pellets were resuspended in 0.1 M tris-HCl buffer (pH 7.4) containing 0.5 mM dithiothreitol, 0.1 mM EDTA, and 20% glycerol. Microsomal protein concentrations were determined by the Bradford protein assay method, using the Bio-Rad Protein Assay kit (Bio-Rad, Hercules, CA, USA) and bovine serum albumin (BSA; Sigma-Aldrich Inc., St. Louis, MO, USA) as a standard according to the manufacturer’s protocol. The total activity of CYP3A was determined by measuring the amount of formaldehyde formed in the reaction of CYP3A-dependent *N*-demethylation of erythromycin [26]. The reaction system contained 50 mM potassium phosphate buffer (pH = 7.4), 3 mM MgCl2 (Fluka, Buches, Switzerland), 12.5 mM erythromycin (Sigma-Aldrich, St-Louis, MO, USA), and 0.5–1 mg microsomal protein. The reaction was started with 0.25 mM NADP (Merck, Darmstadt, Germany) and the samples were then kept on ice. Samples were then centrifuged after addition of 2 mL of 15% trichloroacetic acid. Formaldehyde concentration in the supernatant was measured spectrophotometrically (405 nm) using the Nash reagent containing 2 M ammonium acetate, 0.05 M glacial acetic acid, and 0.02 M acetylacetone.

### 4.2. Statistical Analysis

Data were analyzed using SPSS 24.0 (SPSS Inc., Chicago, IL, USA), STATISTICA 10.0 (StstSoft Inc., Tulsa, OK, USA), and MS Excel 2010 (Microsoft Inc., Redmond Wash, WA, USA). Quantitative data are presented as mean ± SD. After the Shapiro–Wilk test revealed a normal distribution, a one-tailed ANOVA was performed using Tukey’s post hoc tests.

### 4.3. Molecular Docking

The high-resolution (1.70 Å) 3D structure of human CYP3A4 (CYP) was taken from the RCSB protein database [29]. CYP, a member of the oxidoreductase family, has a protoporphyrin IX with iron (HEM) in the catalytic pocket of the enzyme (PDB ID: 5VCC). Only the A chain of the protein and HEM were retained, while water and other small molecules were removed. The initial geometries of the resveratrol-human CYP3A4 complex (RES:CYP), cortisol-human CYP3A4 complex (COL:CYP), and corticosterone-human CYP3A4 complex (CNE:CYP) required for molecular dynamics simulation, was determined by a molecular-docking experiment. The 3D coordinates of resveratrol (RES), cortisol (COL), and corticosterone (CNE) were downloaded from PubChem [30] and converted to pdbqt format using the AutoDockTools 4 Python script prepare_ligand4.py [31]. The CYP was prepared using Chimera 1.14 [32]. The prepared receptor was saved as a pdbqt file after retaining polar hydrogens and adding partial charges (Gasteiger) to each atom. The iron atom of the prosthetic group HEM was set as the center of the grid box with Cartesian coordinates −21.7, −28.1, and −12.4, occupying a total volume of 35 × 35 × 35 Å^3^. The exhaustiveness and the number of modes were set to 100. All docked conformations within a threshold of 4 kcal mol^−1^ relative to the highest score were saved. After visual inspection of the docked poses, the conformation with the best docking score was retained. AutoDock Vina [31] was used for molecular docking simulations.

### 4.4. Molecular Dynamics Simulations

The initial structures of the RES:CYP, COL:CYP, and CNE:CYP complexes for the MD simulation were obtained as a docked resveratrol, cortisol, and corticosterone poses, respectively, with the best docking score. Ligands (RES, COL, and CNE) were parameterized using the AMBER 20 *Antechamber* module [33] and the GAFF [34] force field. The force-field parameters for HEM were taken from [35]. The AMBER ff19SB [36] force field was used for the protein. The protonation state of the titratable residues was determined using the PDB2PQR web-server [37]. The resulting complex was solvated in a truncated octahedral periodic box of TIP3P water molecules. The minimum distance between the atoms of the complexes and the edges of the box was 12 Å. To obtain an electrically neutral system, five Cl^-^ anions were added, followed by the addition of Na^+^ and Cl^−^ ions according to the recommendations of Machado and Pantano [38] to achieve a neutral environment with a salt concentration of 0.15 M.

We followed the MD simulation protocol successfully used to study the binding of resveratrol to 11β-hydroxysteroid dehydrogenase type 1 [11]. Briefly, the geometry was optimized in 10,000 optimization cycles (4000 steepest descent + 6000 conjugate gradient). During the optimization step, both CYP and ligands were constrained with harmonic potential *k* = 10.0 mol^−1^ Å^−2^. The optimized system was heated stepwise from 0 K to 310 K in 500 ps without any constraints. The equilibration phase lasted 500 ps, followed by a productive unconstrained molecular dynamics (MD) simulation of 1 μs. Periodic boundary conditions in all directions were used. Hydrogen atoms were bound using the algorithm SHAKE [10], and the time step was set to 2 fs. Both the pressure (1 atm) and temperature (310 K) were kept constant. The Langevin thermostat controlled the temperature with a collision frequency of 1 ps^−1^. Calculation of the non-bonding interactions was truncated for distances greater than 11 Å. Electrostatic interactions were treated using the particle mesh Ewald method [39]. MD simulations were performed on the Isabella cluster of University Computing Center of University of Zagreb, Croatia, using the molecular dynamics package Amber [40].

### 4.5. Binding Free Energy Calculation

The molecular mechanics/generalized Born surface area (MM/GBSA) protocol [41] was used to estimate the binding free energy (Δ*G_bind_*) between the ligands and CYP. The formula
(3)$$\Delta G_{bind}=\Delta H-T\Delta S\approx \Delta E_{MM}+\Delta G_{sol}-T\Delta S$$
(4)$$\Delta E_{MM}=\Delta E_{internal}+\Delta E_{electrostatic}+\Delta E_{vdW}$$
(5)$$\Delta G_{sol}=\Delta G_{GB}+\Delta G_{SA}$$
was implemented in the MMPSBA.py script of the AmberTools package within the single-trajectory approach. Δ*E_MM_* is the change in MM energy contribution in the gas phase and can be considered as the sum of internal (Δ*E_internal_*), electrostatic (Δ*E_electrostatic_*), and van der Waals (Δ*E_vdW_*) energies. A change in solvation free energy, Δ*G_s__ol_*, has a polar (Δ*G_G__B_*, electrostatic solvation energy) and nonpolar, nonelectrostatic solvation contribution (Δ*G_SA_*). –*T*Δ*S* stands for the conformational entropy upon binding.

The 1 μs trajectories were divided into 20 segments of 50 ns length. To calculate Δ*G_bind_*, 100 snapshots from each segment were sampled at regular time steps. The reported Δ*G_bind_* is the mean ± standard deviation for all 20 segments. The MM/GBSA free energies of binding were decomposed on a per-residue contribution, allowing us to identify the most important interactions among the residues [42]. Due to the high computational cost of calculating the entropy contribution, the entropy term was evaluated for 5 uniformly distributed structures for each segment.

### 4.6. Dynamics Cross-Correlation Map (DCCM) Analysis

The correlated atomic motions of the RES:CYP complex were calculated using the DCCM approach [43,44] as implemented in the pytraj trajectory analysis module of Amber [45]. The elements *C_ij_* of the covariance matrix are calculated as
(6)$$C_{ij}=\frac{{\langle r}_{i}r_{j}\rangle -\langle r_{i}\rangle \langle r_{j}\rangle }{{\left[\left(\langle r_{i}^{2}\rangle -\langle r_{i}\rangle {}^{2})(\langle r_{j}^{2}\rangle -\langle r_{j}\rangle {}^{2}\right)\right]}^{1/2}}$$
where **r***_i_* and **r***_j_* are the position vectors of two atoms *i* and *j*, respectively. The square brackets denote the time averages over the entire trajectory. A contour plot of the matrix C*_ij_* was created in Python 3 using the library seaborn [46].

### 4.7. Cluster Analysis

The geometries of the RES:CYP complex were divided into three clusters using the *k*-means algorithm based on the RMSD of the Cα atoms of each residue. The maximum number of iterations was set to 1000, with the initial number of points randomized and the sieving set to 10. The frames closest to the cluster centers served as representative structures of the identified conformations. The CPPTRAJ module [45] was used to perform cluster analysis.

## 5. Conclusions

The results of the in silico and in vivo studies can be summarized as follows. The in silico data indicate the ability of resveratrol to bind directly within the CYP3A protein catalytic pocket, but with significantly lower binding potential compared to corticosterone and cortisol. In experimental PTSD, resveratrol at a dose of 40 mg/kg prevented the development of behavioral disturbances in 63% of stressed rats. At the same time, the effects of PTSD on tissue metabolism of glucocorticoids were prevented. Simultaneously, the corticosterone concentration in the blood of the stressed rats was restored. In 37% of the stressed rats, resveratrol failed to prevent the behavioral disturbances and abnormalities of tissue glucocorticoid metabolism characteristic of PTSD. The results of the study suggest that the use of resveratrol for pharmacological correction of PTSD is promising. However, the potential side effects should not be ignored.

## Data Availability

The computational data presented in this study are openly available in BIOTECHRI repository at https://urn.nsk.hr/urn:nbn:hr:193:361490 (accessed on 22 May 2023).

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
