# Peer review of "Resistance to Resveratrol Treatment in Experimental PTSD Is Associated with Abnormalities in Hepatic Metabolism of Glucocorticoids"

_ijms, 2023, doi:10.3390/ijms24119333_

Round 1

Reviewer 1 Report

PTSD is a common disease. In this report, the authors focused on PTSD and evaluated the effect of trans-resveratrol on PTSD with an in vivo rat model. The authors also did lots of molecular docking work to further illustrate the mechanism of action. Overall, the paper is well-written and organized,  the manuscript should be published after minor revision.

1.      Table 1, compared to the control group, the PTSD group rats had high numbers of entries to the closed arm but shorter time stay in the closed arm. It doesn’t make sense. Did the author miswrite some data in table 1.

2.      Figure 1, the y axis, the correct unit of concentration should be nM.

3.      What is “RES:CYS” in part 2.5? Should be CYP?

4.      Part 4.1, 12.5 mM/L should be 12.5 mM.

5.      Scheme 1, for better reading, please unify the structures using the same size and fix the bond length.

Reviewer 2 Report

Critique-ijms-2401520-v1

Overall impression.  This study deals with the mechanism by which all-trans resveratrol can improve PTSD in some rats, while aggravating anxiety-like behavior in other rats. It appears that resveratrol has these effects by altering the amount of plasma corticosterone.  The hepatic enzymes 11b-HSD-1 activity was suppressed leading to an increase in CYP3A in resveratrol sensitive rats, while in resistant rats, the activities of both of these enzymes were suppressed

The study was well designed and new knowledge of the effect of all-trans resveratrol on hepatic metabolism was obtained.  However, I would like to see the authors discuss more fully what the future studies on use of all trans resveratrol for treatment of PTSD should be.  Do the authors recommend moving forward with a small clinical trial, or is more work on the rat model needed?  Is there a way to counteract the potential side effects of resveratrol?

Author Response

Overall impression. This study deals with the mechanism by which all-trans resveratrol can improve PTSD in some rats, while aggravating anxiety-like behavior in other rats. It appears that resveratrol has these effects by altering the amount of plasma corticosterone.  The hepatic enzymes 11b-HSD-1 activity was suppressed leading to an increase in CYP3A in resveratrol sensitive rats, while in resistant rats, the activities of both of these enzymes were suppressed.

The study was well designed and new knowledge of the effect of all-trans resveratrol on hepatic metabolism was obtained. However, I would like to see the authors discuss more fully what the future studies on use of all trans resveratrol for treatment of PTSD should be. Do the authors recommend moving forward with a small clinical trial, or is more work on the rat model needed?  Is there a way to counteract the potential side effects of resveratrol?

Answer: We addressed reviewer’s comments in new paragraph in the Discussion section.

Action: The side effects of resveratrol at the behavioral level are presented. Before starting clinical trials, it is necessary to determine the specifics of the side effects of resveratrol in relation to the internal organs. First of all, it is necessary to clarify to what extent oxidative stress is involved in the development of side effects. All this requires additional experimental studies. Another problem is to find ways to eliminate the side effects of resveratrol. In this direction, it seems promising to us to administer resveratrol together with probiotics.

Reviewer 3 Report

On request of IJMS, I have revised the manuscript titled “Resistance to resveratrol treatment in experimental PTSD associated with abnormalities in hepatic metabolism of glucocorticoids”, by Vadim et al.

In this work, authors, using trans- resveratrol (RES), a natural polyphenol extensively studied for its anti-psychiatric effects including protective action in relation to post-traumatic stress disorder (PTSD), have first divided the population of PTSD rats into two phenotypes, treatment-sensitive rats (TSR), and treatment-resistant rats (TRR). They observed that, while in TSR rats, whose anxiety-like behavior was ameliorated by RES, hepatic 11β-HSD-1 activity was suppressed, with a concomitant increase in CYP3A activity, in TRR rats, whose anxiety-like behavior was aggravated by RES, the activities of both enzymes were suppressed. Collectively, the authors demonstrated that resistance of PTSD rats to RES treatment is due to abnormalities in hepatic metabolism of glucocorticoids. They verified that RES inhibits the activity of both 11β-HSD-1 and CYP3A, but that its inhibitory effects in TSR rats are less pronounced than in TRR ones, thus allowing a complete recovery of corticosterone levels, and the activation of CYP3A. Additionally, by studying the free energy of binding of RES, cortisol, and corticosterone to the human CYP3A protein using the molecular mechanics Poisson-Boltzmann surface area approach, they found that RES could affect CYP3A activity but in minor extent than corticosterone and cortisol.

General Comments

Similarly, to what reported by the same authors in their recent article (Ref.9) in which Novak, J.; Tseilikman, V.E.; Tseilikman, O.B.; Lazuko, S.S.; Belyeva, L.E.; Rahmani, A.; Fedotova, J., by an in silico and in vivo study investigated the RES influence on the activity of 11-β-Hydroxysteroid dehydrogenase type 1, here they have investigated the influence of RES on the activity of CYP3A. Anyway, for the first time, the effect of RES on the catalytic activity of CYP3A isoforms toward glucocorticoids, as well as the effect of RES on the relationship between the 11β-HSD-1 and CYP3A pathways of glucocorticoid metabolism in PTSD, have been investigated. Overall, this study evidence that the use of RES for pharmacological correction of PTSD is promising, but the potential side effects should not be ignored.

The present work is promising for publication after the due corrections and improvements.

Following my opinion and suggestions.

1.       The first letters of all word in the title, in headings and sub-headings should be in capital letters.

2.       Check carefully all the manuscript to assure that all abbreviations have been specified at their first mention.

3.       Check the references in the text. They are reported in an incorrect way. The number in square brackets should be before the dot and not after.

4.       Introduction needs improvements. Authors should provide here the chemical structure of RES.

5.       The types of plants containing RES should be indicated, with the related reference.

6.       The main activities of RES and some related case study should be included.

7.       Line 72. Please, correct “are” with “is”.

8.       Line 86. What is EPM?

9.       Line 87. What is PS?

10.   Line 95. Please, specify M±SD.

11.   Table x. in the Table titles and Figure x. in Figure captions should be in bold.

12.   Line 97. Please, insert the due space after line 97.

13.   Table 1. Third column, rows 6 and 7. There is one asterisk (*), which is not reported in the footnotes.

14.   Figure 1 and Figure 2. In the captions, “respective” and “respectively” are used randomly. Please, uniform. Additionally, check carefully the punctuation.

15.   Please, to indicate the Figures of Supplementary file, use Figure S1, S2, S3 etc.

16.   Please, check the template provided by IJMS, and correct the Tables format accordingly.

17.   Lines 302-303. Please, remove the repetition “with concomitant decrease”.

18.   Line 314. Please, use “could” in place of “can”.

19.   Pages 12. Please, remove the structure of RES, which should have been included in the Introduction, as suggested previously.

20.   Line 447. “Scheme” is incorrect, use Figure or Chart. Scheme is used to indicate a reaction procedure.

21.   Line 524. Please, remove the reference 46. References are not allowed in the Supplementary Materials list.

22.   Please, check the layout of the references list used in the template and correct your one accordingly.

Minor editing of English language required

Author Response

On request of IJMS, I have revised the manuscript titled “Resistance to resveratrol treatment in experimental PTSD associated with abnormalities in hepatic metabolism of glucocorticoids”, by Vadim et al.

In this work, authors, using trans-resveratrol (RES), a natural polyphenol extensively studied for its anti-psychiatric effects including protective action in relation to post-traumatic stress disorder (PTSD), have first divided the population of PTSD rats into two phenotypes, treatment-sensitive rats (TSR), and treatment-resistant rats (TRR). They observed that, while in TSR rats, whose anxiety-like behavior was ameliorated by RES, hepatic 11β-HSD-1 activity was suppressed, with a concomitant increase in CYP3A activity, in TRR rats, whose anxiety-like behavior was aggravated by RES, the activities of both enzymes were suppressed. Collectively, the authors demonstrated that resistance of PTSD rats to RES treatment is due to abnormalities in hepatic metabolism of glucocorticoids. They verified that RES inhibits the activity of both 11β-HSD-1 and CYP3A, but that its inhibitory effects in TSR rats are less pronounced than in TRR ones, thus allowing a complete recovery of corticosterone levels, and the activation of CYP3A. Additionally, by studying the free energy of binding of RES, cortisol, and corticosterone to the human CYP3A protein using the molecular mechanics Poisson-Boltzmann surface area approach, they found that RES could affect CYP3A activity but in minor extent than corticosterone and cortisol.

General Comments

Similarly, to what reported by the same authors in their recent article (Ref.9) in which Novak, J.; Tseilikman, V.E.; Tseilikman, O.B.; Lazuko, S.S.; Belyeva, L.E.; Rahmani, A.; Fedotova, J., by an in silico and in vivo study investigated the RES influence on the activity of 11-β-Hydroxysteroid dehydrogenase type 1, here they have investigated the influence of RES on the activity of CYP3A. Anyway, for the first time, the effect of RES on the catalytic activity of CYP3A isoforms toward glucocorticoids, as well as the effect of RES on the relationship between the 11β-HSD-1 and CYP3A pathways of glucocorticoid metabolism in PTSD, have been investigated. Overall, this study evidence that the use of RES for pharmacological correction of PTSD is promising, but the potential side effects should not be ignored.

The present work is promising for publication after the due corrections and improvements.

Following my opinion and suggestions.

  1. The first letters of all word in the title, in headings and sub-headings should be in capital letters.

Answer: Thank you very much for pointing that out. We have corrected the title and all headings and subheadings.

Action: Resistance to Resveratrol Treatment in Experimental PTSD Associated with Abnormalities in Hepatic Metabolism of Glucocorticoids

2.5. Molecular Docking and Molecular Dynamics Simulations

2.6. Identification of Three Conformations

2.7. Binding Free Energy Calculation

4.3. Molecular Docking

4.4. Molecular Dynamics Simulations

4.5. Binding Free Energy Calculation

4.6. Dynamics Cross-Correlation Map (DCCM) Analysis

4.7. Cluster Analysis

  1. Check carefully all the manuscript to assure that all abbreviations have been specified at their first mention.

Answer: We have given the full name of all abbreviations at the first occurrence.

  1. Check the references in the text. They are reported in an incorrect way. The number in square brackets should be before the dot and not after.

Answer: We have corrected the references according to the reviewer's suggestion.

  1. Introduction needs improvements. Authors should provide here the chemical structure of RES.

Answer: As suggested by the reviewer, we have improved the introduction and moved the structures from the Methods section to the Introduction.

Action: In this context, our attention has been drawn to resveratrol, known as (3,5,4-trihydroxy-trans-stilbene (Chart 1), which is a type of biological polyphenols obtained mainly from peanuts, grapes and mulberries. [7] Due to its strong antioxidant action, resveratrol is effective in cardiovascular diseases, type 2 diabetes mellitus, immunodeficiency states, oncology, and behavioral disorders. [8]

RES

COL

CNE

Chart 1. Two‐dimensional structures of trans‐resveratrol (RES), cortisol (COL) and corticosterone (CNE).

  1. The types of plants containing RES should be indicated, with the related reference.

Answer: We have included plants containing RES, with related reference.

Action: In this context, our attention has been drawn to resveratrol, known as (3,5,4-trihydroxy-trans-stilbene (Chart 1), which is a type of biological polyphenols obtained mainly from peanuts, grapes and mulberries. [7]

  1. The main activities of RES and some related case study should be included.

Answer: We have included suggested activities of RES.

Action: Due to its strong antioxidant action, resveratrol is effective in cardiovascular diseases, type 2 diabetes mellitus, immunodeficiency states, oncology, and behavioral disorders. [8]

  1. Line 72. Please, correct “are” with “is”.

Answer: Thank you for pointing that out. It is a typo. We have replaced it.

Action: GC is metabolized by 11-beta-hydroxysteroid dehydrogenase type 1 or 2 (11β-HSD-1 or 11β-HSD-2) and isoforms of cytochrome P450 belonging to the CYP3A subfamily.

  1. Line 86. What is EPM?

Answer: EPM stand for elevated-plus-maze. The abbreviation is explained in the first line of 2.1. paragraph.

  1. Line 87. What is PS?

Answer: PS is predator stress. We have replaced the abbreviation with the full sentence.

Action: We exposed rats to predator stress for 10 days.

  1. Line 95. Please, specify M±SD.

Answer: We have replaced M±SD with a full explanation.

Action: All values are expressed as mean ± standard deviation.

  1. Table x. in the Table titles and Figure x. in Figure captions should be in bold.

Answer: All Tables/Figures in captions are bold.

  1. Line 97. Please, insert the due space after line 97.

Answer: We have inserted due space.

  1. Table 1. Third column, rows 6 and 7. There is one asterisk (*), which is not reported in the footnotes.

Answer: Thank you for pointing that out. We have added explanation of one asterix (*) in the footnotes.

Action: * p < 0.05 respective, control vs. PTSD; *** p < 0.001 respective, control vs. PTSD; ## p < 0.01 respective, PTSD vs. PTSD + TSR, ++ p < 0.01 respective TSR vs TRR.

  1. Figure 1 and Figure 2. In the captions, “respective” and “respectively” are used randomly. Please, uniform. Additionally, check carefully the punctuation.

Answer: We changed all “respectively” to “respective” and checked the punctuation.

Action: Significant values are expressed as * p < 0.05, respective, control vs. TSR and, respective, control vs. PTSD; ### p < 0.01 respective, PTSD vs. TSR, ++ p < 0.01 respective TSR vs. TRR.

  1. Please, to indicate the Figures of Supplementary file, use Figure S1, S2, S3 etc.

Answer: All figures and tables in the supplementary file have been renamed according to the reviewer's suggestion. The necessary changes were also made in the manuscript.

  1. Please, check the template provided by IJMS, and correct the Tables format accordingly.

Answer: The format of the tables has been changed to conform to IJMS template guidelines.

  1. Lines 302-303. Please, remove the repetition “with concomitant decrease”.

Answer: We removed it.

Action: For example, PTSD-susceptible rats were characterized by decreased levels of dopamine with concomitant decreases in the expression of MAO-A and in brain derived neurotrophin factor (BDNF) levels [22].

  1. Line 314. Please, use “could” in place of “can”.

Answer: Thank you, we replace “can” with “could”.

Action: Furthermore, this approach could be successfully applied to the analysis of other pharmacological drugs used to treat both PTSD and anxiety disorders in the future.

  1. Pages 12. Please, remove the structure of RES, which should have been included in the Introduction, as suggested previously.

Answer: As suggested by the reviewer, we have renamed Scheme 1 and moved it to the Introduction.

  1. Line 447. “Scheme” is incorrect, use Figure or Chart. Scheme is used to indicate a reaction procedure.

Answer: As suggested by the reviewer, we have renamed Scheme 1 to Chart 1.

Action: Chart 1. Two‐dimensional structures of trans‐resveratrol (RES), cortisol (COL) and corticosterone (CNE).

  1. Line 524. Please, remove the reference 46. References are not allowed in the Supplementary Materials list.

Answer: In IJMS template is written 'Citations and references in the Supplementary Materials are permitted provided that they also appear in the reference list here.' That is the reason why we believe that reference 46 should not to be removed.

  1. Please, check the layout of the references list used in the template and correct your one accordingly.

Answer: References were handled by Mendeley Reference Manager, and as citation style ‘International Journal of Molecular Sciences’ was used. References style was changed to IJMS template ‘References’ style.

Round 2

Reviewer 3 Report

The work made by authors is satisfying.